# CW nnU-Net for Universal Lesion Segmentation Challenge on 3D Computed Tomography

**Ching-Wei Wang**[*]                              CWEIWANG@MAIL.NTUST.EDU.TW
**Ting-Sheng Su**                                  M11223101@MAIL.NTUST.EDU.TW
**Hikam Muzakkyi**                                 M11123801@MAIL.NTUST.EDU.TW
**Yu-Ching Lee**                                   D10522201@MAIL.NTUST.EDU.TW
*Graduate Institute of Biomedical Engineering, National Taiwan University of Science and Technology, Taipei, Taiwan*

**Editors:** Under Review for MIDL 2024

## Abstract

3D lesion segmentation of oncological computed tomography (CT) is a crucial step in precisely monitoring changes in lesion/tumor growth, which enables the extraction of meaningful information from medical images, aiding in diagnosis, treatment planning and monitoring of diseases. In this research, we developed a highly efficient and effective CW nnU-Net and ensemble models for 3D lesion segmentation on CT for the Universal Lesion Segmentation (ULS) Challenge, which will be held jointly with 2024 medical imaging with deep learning (MIDL) conference at Paris, France. The proposed approach was built with a reasonably cheap Nvidia RTX 4080 GPU card and outperformed the baseline models in both development and test phase. In the final test phase, the proposed model ranks as the **3rd place** among **632 participants worldwide** (accessed date: 2024/4/29), achieving a Challenge Score of 0.73, Segmentation DICE of 0.70 and Consistency DICE of 0.79. In the development phase, the proposed CW nnU-Net achieved a Challenge Score of 0.81, Segmentation DICE of 0.78 and Consistency DICE of 0.92. For computational efficiency, CW nnU-Net takes only 3.25s for processing each VOI on the Grand Challenge platform server with a single T4 GPU and less than 2s using a local PC with RTX4080.

**Keywords:** Deep Learning, Oncological computed tomography, 3D Lesion Segmentation.

## 1. Introduction

3D lesion segmentation of oncological computed tomography (CT) is crucial in precisely monitoring changes in lesion/tumor growth, which enables the extraction of meaningful information from medical images, aiding in diagnosis, treatment planning and monitoring of diseases. Automatic lesion segmentation on CT scans offers advantages over manual segmentation, including improved efficiency, reproducibility, accuracy, and standardization, enabling more precise quantitative analysis and facilitating the translation of research into clinical practice. However, there are still challenges in developing and deploying these techniques, including the need for large and diverse datasets, the risk of overfitting, and the need for robust validation and evaluation methods. The Universal Lesion Segmentation (ULS) challenge provides annotated lesion data in the form of volumes of interest (VOIs) (de Grauw et al., 2023) for 3D lesion segmentation of 3D CT, containing various types of

---

[*] Corresponding Author

lesions such as abdominal lesions, bone lesions, pancreas lesions, kidney lesions, liver lesions, lung nodules, lung lesions, colon lesions, lymph nodes, mediastinal and assorted lesions (see Table 1). In this research, we developed a CW nnU-Net and ensemble methods for 3D lesion segmentation on CT for the ULS Challenge.

Table 1: Materials: ULS Challenge Dataset.

| Training (Public) | Validation (Private) | Testing (Private) |
|---|---|---|
| fully annotated 6514 lesions partially annotated 33060 lesions | 50 lesions / 16 patients (from 2 centers (25 each)) | 725 lesions / 268 patients |

## 2. Method: CW nnU-Net

### 2.1. Data Pre-processing

No data pre-processing or resampling is performed for high computational efficiency and better efficacy. In addition, no data augmentation is applied due to limited time.

### 2.2. Model Architecture

The proposed CW nnU-Net is based on a U-Net architecture with a residual encoder, consisting of a 7-stage encoder with a series of convolutions per stage, (1,3,4,6,6,6,6) respectively, enhancing the model's depth and complexity at deeper stages and a 6-stage decoder with one convolution per stage, (1,1,1,1,1,1) respectively, to reduce the computational load while maintaining performance. In comparison to the standard nnUnet architecture (1,3,4,6,6,6) followed by (1,1,1,1,1), we devised our own configuration with a deeper architecture (1,3,4,6,6,6,6) followed by (1,1,1,1,1,1) decoder by adding one additional stage where (1,3,4,6,6,6,6) specifies the number of convolutions per stage. The loss function includes cross-entropy and Dice loss.There are 710,279,660 parameters in our network, and the total model size is 1.52 GB.

### 2.3. Training Strategy

For data selection, we excluded 138 normal volumes without lesion masks from the fully annotated set and utilized only the remaining 6376 fully annotated lesions for training. Experiments were conducted on lesions of size $1 \times 128 \times 256 \times 256$ with batch size equal to 1 due to usage of a single cheap GPU card (RTX 4080). In addition, the weights of the baseline model is adopted as the pretrained weights for continuous training. In analyzing the two baseline models provided by the challenge organizers, we identified a key difference in the '$unet\_max\_num\_features$' parameter, with values set at 320 and 384, respectively, and we found that the model with the parameter value of 384 exhibited a larger size and higher performance. Hence, in training, we utilized the model parameter value of 384. Table 2 compares the proposed models with the baseline models.

Table 2: Model Comparison.

| Model | Batch Size | unet_max_num_features | Initial Learning Rate | Epoch | Model Size |
|---|---|---|---|---|---|
| Best CW nnU-Net | 1 | 384 | 3e-05 | 500∼1000 | 1.52 GB |
| 2nd Best CW nnU-Net | 1 | 384 | 1e-05 | 500∼1500 | 1.52 GB |
| ULS BaseLine (Model 901) | 3 | 384 | 0.0025 | 0∼500 | 1.52 GB |
| ULS BaseLine-s (Model 400) | 2 | 320 | 0.0025 | 0∼500 | 1.13 GB |

Table 3: Final Test Phase Result in the Leaderboard (out of 632 participants).

| Rank | Model | Challenge Score | Segmentation DICE | Long-Axis SMAPE | Short-Axis SMAPE | Consistency DICE |
|---|---|---|---|---|---|---|
| 3rd | Best CW nnU-Net | **0.7297** | **0.7033 ± 0.2398** | **0.1095** | 0.1197 | 0.7854 ± 0.2547 |
| 4th | ULS BaseLine | 0.7294 | 0.7028 ± 0.2395 | 0.1116 | **0.1195** | **0.7872 ± 0.2522** |

## 2.4. Ensemble Models

Ensemble models were built based on our previous efforts (Wang et al., 2017, 2023a,b). A multiprocessing approach is built for the computational efficiency to ensure that a model could perform inference on a stack of 100 3D VOIs within 9 minutes.

## 3. Result

In the final test phase (see Table 3), the proposed CW nnU-Net ranks as the **3rd place** among **632 participants** worldwide (accessed date: 2024/4/29) and outperforms the baseline model, achieving Challenge Score: 0.73, Segmentation DICE: 0.70 and Consistency DICE: 0.79. For computational efficiency, CW nnU-Net takes only 3.25s for processing each VOI on the Grand Challenge platform server with a single T4 GPU and less than 2s using a local PC with RTX4080. During the development phase, we concurrently built ensemble models based on two baselines. Table 4 shows that the ensemble model by conjunction of two baselines outperforms individual baselines. Furthermore, both CW nnU-Net models perform better than the baseline models, and the CW nnU-Net model with higher learning rate performs slightly better than the one with lower learning rate, which might get stuck at a local minimum. It would be interested in evaluating the ensemble of the two best CW nnU-Nets.

Table 4: Development Phase Result in the Leader board (out of 632 participants).

| Rank | Model | Challenge Score | Segmentation DICE | Long-Axis SMAPE | Short-Axis SMAPE | Consistency DICE |
|---|---|---|---|---|---|---|
| 5th | Best CW nnU-Net | **0.8128** | 0.7845 ± 0.1613 | 0.0597 | 0.0832 | **0.9235 ± 0.0530** |
| 6th | Ensemble BaseLine (Conjunction) | 0.8125 | **0.7863 ± 0.1660** | 0.0615 | 0.0840 | 0.9072 ± 0.0769 |
| 8th | 2nd Best CW nnU-Net | 0.8119 | 0.7838 ± 0.1588 | 0.0605 | 0.0863 | 0.9223 ± 0.0531 |
| 9th | ULS BaseLine | 0.8119 | 0.7836 ± 0.1560 | 0.0607 | **0.0816** | 0.9217 ± 0.0575 |
| 19th | ULS BaseLine-s | 0.8101 | 0.7845 ± 0.1656 | **0.0565** | 0.0862 | 0.8966 ± 0.0768 |
| 21th | Ensemble BaseLine (Disjunction) | 0.8099 | 0.7818 ± 0.1550 | 0.0567 | 0.0891 | 0.9173 ± 0.0586 |

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
