# OpenReview forum: "CW nnU-Net for Universal Lesion Segmentation Challenge on 3D Computed Tomography"
_MIDL.io/2024/Short_Papers — MIDL 2024 Short Papers_

### Official Review · Reviewer_bLxP · 2024-04-15

**Confidence:** 4
**Final Rating:** 3.5

**Review:**

This work describes a method submited to the ULS23 challenge, which finished in the 3rd place during the official test-phase of the challenge (ahead of the 4th ranking that was the ULS-baseline model made by the organizers).

Strenghts:
- 3rd best method in the test-phase of the ULS challenge (offering small improvement as per the paper over the 4th method, which was ULS baseline made by the organizers)

Weaknesses:
- Not much technically new insights. The model is based on nnUnet. To my understanding, the method is the model (nnUnet) provided by organizers pretrained as baseline, and finetuned further with a different learning rate, offering a small improvement.
- Some information need further clarification.

Comments to fix (eg for camera ready) in case the paper is accepted for publication:
- I dont think the phrase "and late participation for us to complete the challenge task within one month as we only started exploring this challenge from mid March, 2024." adds useful information to reader.
- "[1,3,4,6,6,6,6]" is a bit ambiguous what this means. I believe it means how many convolutions per "stage" of unet. Please rephrase to make it more appropriately, e.g. "per stage, (1,3,4,6,6,6,6) respectively.".
- Change the text to state/clarify if the given architecture [1,3,4,6,6,6,6] followed by [1,1,1,1,1,1] decoder, is the standard of nnUnet or this number of convs is your own configuration.
- "A multiprocessing approach is built for the computational efficiency." => Unclear what this means.
- Why do the authors claim there were 577 participants? The leaderboard only shows 76 that submitted to the development challenge (https://uls23.grand-challenge.org/evaluation/development-phase-leaderboard/leaderboard/). 500+ shown as "participants" on the website's map are probably the amount of people that signed up, e.g. to download the data? But not those that actively participated? Those that actively participated are 76, that submitted something to the leaderboard, even just in the development stage. I would strongly recommend you to change the number in your paper, as 76 seems the more representative number to me, as it's the number of methods actually compared with yours, while the 500+ never submitted/compared (perhaps just downloaded the data).
- Table 3, do you have info to add about the Rank 1 and 2 methods? If yes, it would make the. table far more informative.
- "Conjunction" and "Disjunction" about the ensembles are not iinterpretable. What do you mean? Rephrase and clarify in text.

Justification of score:
The work does not offer much new technical insights. The method is essentially the ULS baseline given by organizers, further finetuned with different learning rate, and this gave a small improvement over the baseline. Regardless, the presence of the authors, who now have experience on the problem, could contribute to discussions in the conference about the ULS challenge and the problem.

---

### Decision · Program_Chairs · 2024-04-26

Accept